# Learning Graph Cellular Automata

**Daniele Grattarola**
Università della Svizzera italiana
grattd@usi.ch

**Lorenzo Livi**
University of Manitoba

**Cesare Alippi**
Università della Svizzera italiana
Politecnico di Milano

## Abstract

Cellular automata (CA) are a class of computational models that exhibit rich dynamics emerging from the local interaction of cells arranged in a regular lattice. In this work we focus on a generalised version of typical CA, called *graph* cellular automata (GCA), in which the lattice structure is replaced by an arbitrary graph. In particular, we extend previous work that used convolutional neural networks to learn the transition rule of conventional CA and we use graph neural networks to learn a variety of transition rules for GCA. First, we present a general-purpose architecture for learning GCA, and we show that it can represent any arbitrary GCA with finite and discrete state space. Then, we test our approach on three different tasks: 1) learning the transition rule of a GCA on a Voronoi tessellation; 2) imitating the behaviour of a group of flocking agents; 3) learning a rule that converges to a desired target state.

## 1   Introduction

Cellular automata (CA) [1] are discrete computational models that consist of a regular lattice of cells, each associated with a state, and a transition rule. The transition rule is usually defined to update the state of each cell as a function of its current state and the state of its neighbours. Even when using simple transition rules and low-dimensional lattices, CA can exhibit very intricate patterns that emerge from the repeated synchronous application of the transition rule. Despite being almost a century-old idea, many different forms of CA are still studied today in dynamical systems theory [2]. In this paper, we focus on a generalised version of CA called *graph cellular automata* (GCA) [3], which relaxes the assumption of the lattice structure and allows cells to have an arbitrary and variable number of neighbours.

Starting from the observation that GCA are a generalised version of lattice-based CA, in which the underlying lattice is replaced by an arbitrary graph, we focus on the analogous parallel that exists between convolutional neural networks (CNNs) and graph neural networks (GNNs) [4, 5, 6]. Namely, the goal of this paper is to study the application of GNNs to learn a desired GCA transition rule, a setting that we refer to as *Graph Neural Cellular Automata* (GNCA).

The inspiration for this work comes from a recent series of papers that studied how to train a convolutional neural network to approximate a desired CA transition rule, which we extend in two significant ways. First, we present a general-purpose architecture for GNCA and show that it is sufficient to represent any arbitrary GCA with finite and discrete state space. Based on well-known results on the expressive power of neural networks and GNNs, we also propose that the same architecture is expressive enough to represent any transition function on continuous state spaces. Then, we present several classes of problems that can be formulated as computation with GCA. We use our architecture to learn a varied family of GCA with non-trivial features including discrete and continuous state spaces, complex or unknown transition rules, dynamic graphs, and the ability to globally coordinate through local message exchanges. First, we train a GNCA to approximate a GCA defined on the Voronoi tessellation of random points, where the underlying graph is fixed throughout the evolution of the system and the state space is binary. Then, we focus on a dynamic GCA with

35th Conference on Neural Information Processing Systems (NeurIPS 2021).

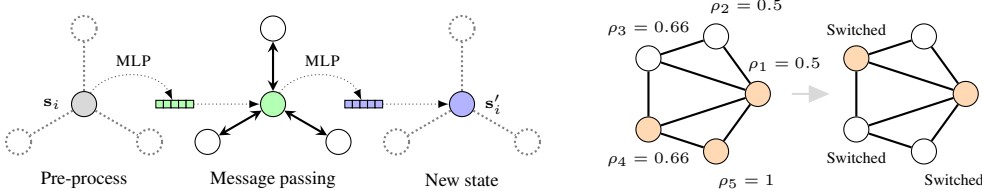

(a) Schematic view of GNCA computation.

(b) Voronoi GCA transition.

Figure 1: (a) Schematic view of the GNCA computation. The central block represents the message-passing function of Eq. (4). (b) Example transition of a Voronoi GCA with binary states (represented by colours) and transition rule as described in Eq. (5) with $\kappa = 0.6$. Cells with a neighbourhood density $\rho_i > \kappa$ switch state.

continuous state space and variable connectivity between cells. We consider the Boids algorithm [7], a Markovian and distributed multi-agent system designed to simulate the flocking of birds, and we train the GNCA to reproduce its behaviour. Finally, we consider the task of learning GNCA that converge to a target state in a continuous state space. Specifically, we train the GNCA to reconstruct the coordinates of a geometric point cloud.

The practical implications of this work are evident when considering that distributed systems are 1) the norm in nature, where complex behaviour is observed as an emergent property of simple and distributed computation, and 2) ubiquitous in technology, with the advent of cyber-physical systems that need to operate and react to local events while coordinating to achieve a target behaviour (for example, self-driving cars that act locally to avoid crashes but must prevent traffic globally). By applying GNNs to model GCA on systems described by graphs, we enable the discovery of unknown rules that allow us to solve tasks through emergent and distributed computation.

## 2 Related works

This work extends the results of many foundational works on learning CA with neural networks, which so far have only focused on regular lattices.

A seminal work on the subject is that of Wulff and Hertz [8], who successfully trained a small neural network (NN) to imitate 1D and 2D binary CA with chaotic or complex dynamics.

A subsequent approach is that of Elmenreich and Fehérvári [9], who designed an evolutionary algorithm to identify a NN transition rule that would generate a desired pattern (a process called *morphogenesis*). Similarly, Nichele et al. [10] used the NEAT genetic algorithm [11] and compositional pattern producing NNs [12] for morphogenesis and pattern replication in a CA with discrete states.

Recently, Gilpin [13] showed how to implement any CA transition rule using the receptive field of a CNN, as well as showing that arbitrary binary rules can be learned end-to-end with gradient descent. In a follow-up analysis, Springer and Kenyon [14] observed that CNNs often fail to converge to exact solutions when trained to reproduce the Game of Life ruleset and that they often require many more parameters than the minimal number necessary to implement the exact transition rule, as predicted by the lottery ticket hypothesis [15]. The idea of learning 2D CA transition rules was also explored by Aach et al. [16], who proposed an encoder-decoder architecture with the ability to generalise to unseen neighbourhood configurations and sizes.

Mordvintsev et al. [17] recently introduced Neural Cellular Automata (NCA). The idea of NCA is to use a CNN to learn the transition rule of a 2D CA with a continuous and multi-dimensional state space, which they interpret as an image. In particular, they trained the NCA to converge to a specific image. They showed that, with proper training, NCA could reconstruct missing patches of the target image and were resilient to many classes of state perturbations. Randazzo et al. [18] also showed that NCA can be used to classify MNIST digits. This is an example of local behaviour leading to a global agreement about the overall structure of the system.

Since a CNN can be interpreted as a particular case of a GNN with fixed-size neighbourhood and anisotropic filters, what we propose here is a general extension of the works mentioned above.

## 3  Method

### 3.1  Graph Cellular Automata

A GCA is a 4-tuple $(\mathcal{G}, \mathcal{S}, \mathcal{N}, \tau)$ where $\mathcal{G} = (V, E)$ is a graph with node set $V$ ($|V| = n$) and edge set $E \in V \times V$, $\mathcal{S}$ is the state set, $\mathcal{N} : V \to 2^V$ is the neighbourhood function s.t. $\mathcal{N}(i) = \{j \mid (j, i) \in E\}$, and $\tau$ is a local transition rule. Let $\mathbf{s}_i \in \mathcal{S}$ indicate the state of node $i$. The transition rule is then defined as

$$\tau(\mathbf{s}_i) : \{\mathbf{s}_i\} \cup \{\mathbf{s}_j \mid j \in \mathcal{N}(i)\} \mapsto \mathbf{s}_i'. \tag{1}$$

For brevity, we denote with $\mathbf{S} \in \mathcal{S}^n$ the state configuration for the entire automaton and we write $\mathbf{S}' = \tau(\mathbf{S})$ to indicate the synchronous application of $\tau$ to all cells.

The original CA formalism allows for transition rules that can treat the different neighbours of a cell differently, according to their position with respect to the cell. This idea is, for instance, at the base of the Wolfram code for enumerating CA transition rules [19]. In GCA a similar anisotropic behaviour can be obtained by uniquely enumerating the neighbours of each cell and adapting the transition rule accordingly. More generally, we can assign to each edge $(i, j) \in E$ an attribute $\mathbf{e}_{ij} \in \mathbb{R}^{d_e}$ encoding, for example, the direction, distance, or unique identification of node $i$ relative to node $j$. It is also possible to consider any arbitrary attributes associated with edges, representing abstract relations between any two neighbouring cells. In short, we can see anisotropy in GCA as a dependence of the transition rule on the specific relation that exists between a cell and a neighbour:

$$\tau(\mathbf{s}_i) : \{\mathbf{s}_i\} \cup \{\mathbf{s}_j, \mathbf{e}_{ji} \mid j \in \mathcal{N}(i)\} \mapsto \mathbf{s}_i'. \tag{2}$$

The usefulness of this definition will become evident in later sections.

### 3.2  Graph Neural Networks

Graph Neural Networks (GNNs) are a family of models designed to process graph-structured data. The core functionality of many GNNs can be described with a message-passing scheme. Let $\mathbf{h}_i \in \mathbb{R}^{d_h}$ represent a vector attribute associated with the $i^{\text{th}}$ node of a graph and $\mathbf{e}_{ij} \in \mathbb{R}^{d_e}$ an edge attribute as defined above. A message-passing GNN is a function that computes a vector $\mathbf{h}_i' \in \mathbb{R}^{d_h'}$ according to the following scheme:

$$\mathbf{h}_i' = \gamma\left(\mathbf{h}_i, \square_{j \in \mathcal{N}(i)}\, \phi\left(\mathbf{h}_i, \mathbf{h}_j, \mathbf{e}_{ji}\right)\right), \tag{3}$$

where $\phi$ is the message function, $\square$ is a permutation-invariant operation to aggregate the set of incoming messages, and $\gamma$ is the update function. When $\phi$, $\square$, and $\gamma$ are differentiable operations, message-passing layers can be stacked and their parameters can be learned end-to-end with backpropagation and stochastic gradient descent.

In this work, specifically, we adopt a message-passing architecture inspired by You et al. [20] s.t.

$$\mathbf{h}_i' = \mathbf{h}_i \,\Big\|\, \sum_{j \in \mathcal{N}(i)} \mathrm{ReLU}(\mathbf{W}\mathbf{h}_j + \mathbf{b}), \tag{4}$$

where $\|$ indicates vector concatenation, ReLU is the Rectified Linear Unit activation, and $\mathbf{W} \in \mathbb{R}^{d_h' \times d_h}$ and $\mathbf{b} \in \mathbb{R}^{d_h'}$ are trainable parameters. The message-passing block is also preceded and followed by multi-layer perceptrons (MLPs) with ReLU activations, whose task is to learn non-relational transformations of the node features [20]. The architecture is schematically shown in Fig. 1a. Note that Eq. (4) can easily be extended to account for edge attributes as well, although we will see in later sections that different use cases may benefit from different ways of incorporating that information. We mention two main techniques to achieve this kind of anisotropy, both based on making the message function $\phi$ edge-dependent: 1) the weights matrix $\mathbf{W}$ can be computed as a function of $\mathbf{e}_{ji}$, through a *kernel generating network* [21], and 2) $\mathbf{e}_{ji}$ can be concatenated to $\mathbf{h}_j$ to compute the messages.

### 3.3  Graph Neural Cellular Automata

The goal of this paper is to explore the parallel that exists between GCA and GNNs. In particular, here we consider a setting in which the state transition function of a generic GCA is implemented with a GNN, which we indicate as a *Graph Neural Cellular Automaton* (GNCA).

The relation between the two models can be immediately seen by comparing Eqs. (2) and (3), as the GNN model contains all the necessary elements to describe a GCA transition rule. First, when the GNN is configured so that $d_h = d'_h$, then both functions are endomorphisms that depend only on node $i$ and its neighbourhood $\mathcal{N}(i)$. Then, by considering $\mathcal{S} \equiv \mathbb{R}^{d_h}$, the state of the GCA can be interpreted as a graph signal s.t. $\mathbf{h}_i = \mathbf{s}_i$. Finally, as previously mentioned, the GNN can be configured to take into account edge attributes, allowing it to model transition rules that can be both isotropic or anisotropic. We also note that CA rules with larger neighbourhoods (*e.g.*, the Lenia CA [22]) can be implemented by our GNCA by stacking multiple message-passing layers or by re-defining the neighbourhood function to include higher-order neighbours. This will be explored in future work.

We indicate with $\tau(\mathbf{S})$ the transition function of a GCA and with $\tau_\theta(\mathbf{S})$ the transition function of a GNCA with learnable parameters $\theta$. The dependency of either function on graph $\mathcal{G}$ is left implicit. Finally, we use $d$ to indicate the states' dimension. The specific GNN architecture of $\tau_\theta$ that we consider in Eq. (4), besides being motivated by the results of You et al. [20], has a sufficiently high number of parameters which, as noted by Springer and Kenyon [14], can be a deciding factor for learning the transition rule of a CA. This requirement is also somewhat reflected by the architectural choices of Mordvintsev et al. [17].

**Implementing GCA with $M$-state space**   One of the main results of Gilpin [13] was to show different ways to implement an arbitrary $M$-state CA with a CNN. The most straightforward implementation proposed by the author relied on two main neural computation blocks:

1. Two convolutional layers with 1x1 filters, to compute a one-hot encoding of the states;
2. One convolutional layer with 3x3 filters, to match all of the possible input patterns of the transition rule and compute the corresponding output.

The general principle used to design the transition rule of a CA with a CNN can be straightforwardly extended to GCA and GNNs. Consider a GCA with state space $\mathcal{S} = \{1, ..., M\}$ and $n$ cells, with maximum node degree $d_{max} \leq n$ (and, usually, $d_{max} \ll n$).

The first block of Gilpin's implementation remains the same, since it is a transformation applied individually to each node, independently of its neighbours. The second block requires us to be more careful. First, we note that Gilpin's implementation covers the most general case in which the transition rule is anisotropic. By generalising this principle to the case of GCA, as discussed in Sec. 3.1, we require that each cell has up to $d_{max}$ neighbours identified by a unique one-hot vector edge attribute. The pattern matching mechanism proposed by Gilpin can then be implemented with a bank of $d_{max}$ weight matrices of shape $M \times M^{d_{max}}$, that is used to compute the message from each neighbour, according to its associated edge attribute (*e.g.*, consider the work of Simonovsky and Komodakis [21]).

We note, however, that this case represents an upper bound in complexity, since many GCA rely on isotropic rules and better implementations can be obtained if the structure of the GCA rule is known. For example, Conway's Game of Life can be implemented with a CNN consisting of one convolutional layer with 3x3x5 filters, with one channel to implement the identity and four channels to count neighbours, and two convolutional layers with 1x1x5 filters for pattern-matching [13]. We note that the GNCA covers this result as it natively implements the identity (the concatenation of Eq. (4) can be replaced with a sum to match the exact implementation) and neighbour counting can be achieved with $\mathbf{W} = [1, 1, 1, 1]$ and $\mathbf{b} = [-1, -2, -3, -4]$ (*cf.* [13, Appendix A]). Finally, the 1x1x5 convolutional layers are equivalent to the post-processing dense layers described in Sec. 3.2.

In Sec. 4.1, we will see a similar minimal implementation for an outer-totalistic GCA.

**Extension to continuous state space**   We can also conjecture that the architecture described in Sec. 3.2 is expressive enough to represent any transition function for a GCA with continuous state space. First, it is a well-known result of machine learning theory that a two-layer MLP, when appropriately configured with a sufficient number of neurons, is a universal approximator for any Borel measurable function. The pre-processing MLP of the general GNCA architecture is therefore sufficient to compute any desired representation of the states. Then, building on the results of Zaheer et al. [23], we can also see that the structure of the GNCA allows us to model any permutation invariant function of the neighbours. Finally, by concatenating the representation of a node to the

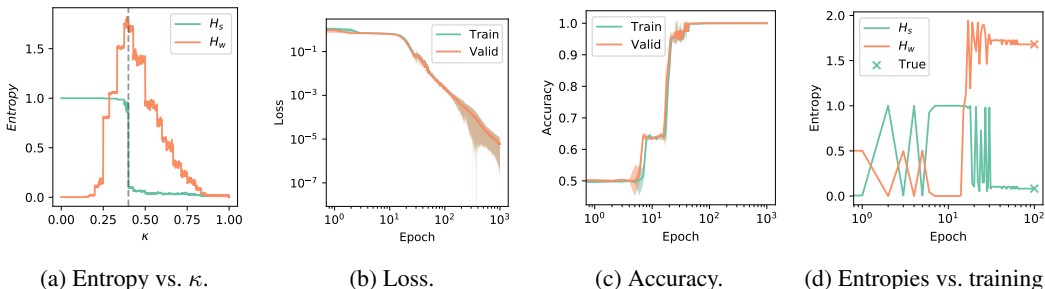

| (a) Entropy vs. $\kappa$. | (b) Loss. | (c) Accuracy. | (d) Entropies vs. training. |

Figure 2: (a) Shannon $H_s$ and word $H_w$ entropy vs. threshold $\kappa$ for the Voronoi GCA. The edge-of-chaos transition is marked at $\kappa = 0.4$. (b) Median training and validation loss of the GNCA. (c) Median training and validation accuracy of the GNCA.[1] (d) Entropy of the GNCA as training progresses. The two "x" indicate the real entropy values of the target Voronoi GCA.

representation of its neighbours and giving the result as input to the post-processing MLP, we are also able to compute any transition to the following state. Of course, these results are conditioned on the two MLPs being sufficiently large, and there are no guarantees that the training algorithm will be able to identify the correct function.

## 4 Experiments

We now consider different experiments aimed at showcasing the capabilities of our GNCA architecture. We take inspiration from the literature on learning lattice-based CA to design three experimental settings with different goals. First, we show a basic application to confirm that the GNCA is indeed capable of learning a desired GCA transition rule via supervised learning. Then, we extend this setting to a more complex case in which the state space is continuous and the underlying graph is dynamic, to demonstrate the applicability of the GNCA to multi-agent systems modelling. Finally, we consider the task of learning an unknown transition rule simply by specifying the target behaviour of the GNCA. Specifically, we extend the work of Mordvintsev et al. [17] to the GCA case and train our GNCA to converge to a desired state. Here we report a summary of the experiments and refer the reader to the supplementary material for more details.

### 4.1 GNCA on Voronoi tessellation

We begin by studying a binary GCA with density-based transition rules and threshold dynamics [24], and for which $\mathcal{G}$ is given by the Delaunay triangulation of a random 2D point cloud. The cells of this type of GCA are the cells of the Voronoi tessellation of the points. Let $\mathcal{S} = \{0, 1\}$ be a binary state space, $\rho_i = \frac{1}{|\mathcal{N}(i)|} \sum_{j \in \mathcal{N}(i)} \mathbf{s}_j$ the neighbourhood density of a node, and $\kappa \in [0, 1]$ a threshold, then the GCA rule is given by:

$$\tau(\mathbf{s}_i) = \begin{cases} \mathbf{s}_i, & \text{if } \rho_i \leq \kappa \\ 1 - \mathbf{s}_i, & \text{if } \rho_i > \kappa \end{cases}. \tag{5}$$

An example transition on such a *Voronoi GCA* is shown in Fig. 1b.

The goal of this experiment is to study the base case in which $\mathcal{G}$ is static and planar, which is the simplest extension of typical outer-totalistic CA (like the Game of Life) to the setting of an irregular arrangement of the cells. Having a binary state space simplifies the problem and allows us to treat this as a node classification task.

**Measures of GCA complexity** Following the method of Marr and Hütt [24], we quantify the complexity of our finite-state GCA using two entropy measures, estimated from the sequences of states observed by repeatedly applying the transition rule. The average Shannon entropy is defined as $H_s = \frac{1}{n} \sum_{i=1}^{n} -p_i^s \log_2 p_i^s$, where $p_i^s$ is the empirical probability that node $i$ is in state $s$. The

---

[1]For Figs. 2b and 2c, the curves have some significant outliers and therefore, instead of the mean and standard deviation, we report the median and the median absolute standard deviation interval over 5 runs.

average word entropy is $H_w = \frac{1}{n}\sum_{i=1}^{n}(\sum_{l=1}^{t} -p_i^l \log_2 p_i^l)$, where $p_i^l$ is the number of constant words of length $l$ (for any state) found in the state sequence of node $i$. When combined, these two measures are informative enough to separate GCA into the four Wolfram classes of CA complexity [24]. As shown in Fig. 2a, the Voronoi GCA considered here has a sharp edge-of-chaos transition [25] at around $\kappa = 0.4$, above which the CA switches to non-chaotic dynamics. This is marked by a sharp decrease in Shannon entropy and by reaching peak word entropy (which indicates the occurrence of interesting patterns of states). In the following experiments, we consider $\kappa = 0.42$ as it exhibits both non-chaotic and non-trivial dynamics.

**Training**  We start by computing $\mathcal{G}$ as the Delaunay triangulation of $n = 1000$ random points sampled uniformly in $[0, 1] \times [0, 1]$. We generate training examples for the model by sampling mini-batches of 32 random binary states $[\mathbf{S}^{(1)}, \dots, \mathbf{S}^{(32)}]$, $\mathbf{S}^{(k)} \in \mathcal{S}^n$, and we train the GNCA by minimising the negative log-likelihood between the true successor states $\tau(\mathbf{S}^{(k)})$ and the predicted next states $\tau_\theta(\mathbf{S}^{(k)})$.

**Results**  Figs. 2b and 2c show the evolution of the GNCA's loss and accuracy on the training and validation sets, averaged over 10 runs. We see that the model quickly converges to a near-perfect approximation of the transition function and that the loss continues to gradually improve even after reaching 100% validation accuracy. Fig. 2d shows the Shannon and word entropies of the model as training progresses. To evaluate the entropy at each training epoch, the GNCA is evolved autonomously (*i.e.*, feeding its predictions back as input) for 1000 steps and the two entropy measures are computed from the observed state trajectories. Note that the GNCA's predictions are rounded to 0 or 1 during inference. This analysis lets us comment on two aspects. First, the GNCA is able to quickly learn a transition function with a very similar entropy to the true transition function (marked with an "x" in the figure), although it takes a while to reach the exact target values. Second, we see that the GNCA eventually learns a robust approximation of the true transition function, which does not diverge from the true trajectory when evolved autonomously.

**Minimal exact implementation**  As discussed in Sec. 3.3, when the form of the transition rule is known, it is often possible to provide a much more compact implementation of the GNCA. For the rule described in Eq. (5), we can see that the GNCA has to implement (or learn to implement) two main operations: calculating the density of alive neighbours and mapping the state to its successor as a function of the state itself, the density, and the threshold $\kappa$. The first step can be achieved with a minor modification to Eq. (4) to use an average aggregation instead of the sum, and by setting $\mathbf{W} = 1$ and $\mathbf{b} = 0$. Alternatively, it is also possible to keep the sum aggregation and set $\mathbf{W} = [1, 0]$, $\mathbf{b} = [0, 1]$ (one neuron to count alive neighbours, and one to count all neighbours), although this requires an extra neuron in the following block. We also remove the pre-processing MLP since the states themselves are enough to compute the required representation. Then, the transition can be computed with the post-processing MLP of the GNCA. As shown in Fig. 3, we see that the transition rule of Eq. (5) is essentially equivalent to the famous XOR problem [26, 27] and can therefore be solved with a MLP with 2 hidden neurons and one output neuron. We empirically trained one such architecture with ReLU hidden activation and sigmoid output (mimicking the structure of the full post-processing MLP described above) and found that one possible solution is given by the following set of weights:

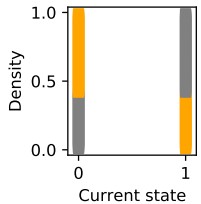

Figure 3: GCA rule of Eq. (5). Colours indicate the output: grey is 0, orange is 1.

$$\mathbf{W}_1 = \begin{bmatrix} -1.98 & 1.64 \\ 2.63 & -2.8 \end{bmatrix}; \quad \mathbf{b}_1 = [-0.46 \quad 0.17]; \quad \mathbf{W}_2 = [3.3 \quad 3.3]; \quad \mathbf{b}_2 = [-2.1].$$

## 4.2 GNCA for agent-based modelling

Our second experiment consists of learning a GCA with continuous state space and in which the connectivity between cells changes over time. Specifically, we take a set of cells whose states represent their positions and velocities on a plane, $\mathcal{G}$ is a dynamic graph given by a fixed-radius neighbourhood of each cell at each step, and the state of the cells is updated with the Boids algorithm [7]. This setting is more general than the previous two experiments, since $\mathcal{G}$ is now a dynamic graph, and possibly a more realistic approximation of many real-world phenomena. We still consider this as a

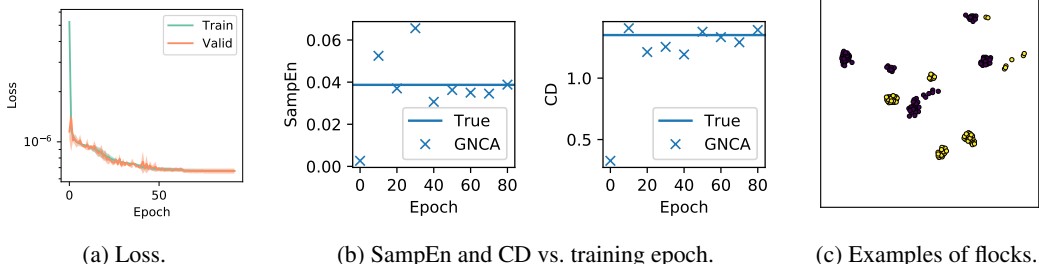

(a) Loss.                    (b) SampEn and CD vs. training epoch.           (c) Examples of flocks.

Figure 4: (a) Average and standard deviation of the training and validation loss of the GNCA (5 runs). (b) SampEn and CD of the GNCA as training progresses. The horizontal lines indicate the true values of the target boids GCA. (c) Examples of flocks forming in the true boids system (purple) and the learned GNCA (yellow).

kind of CA since the characterising feature of CA is that the transition function depends on a cell's neighbours and is Markovian, although this setting is better described as a generic multi-agent system.

The transition function of the Boids algorithm is highly non-linear and is summarised in Algorithm 1 (see the supplementary material for details). Briefly, at each step, the velocity of each boid is updated and its position changes accordingly. The new velocity of each boid is computed as a function of its current velocity and the positions and velocities of the neighbours. Finally, some constraints on the maximum speed and turn angle are enforced, as well as a soft-boundary condition that pushes the boids away from the edges of the simulation box.

**Algorithm 1:** Pseudo-code for Boids [7].

**Input:** $n$: n. of boids; $T$: n. of steps;
1 **for** $i \leftarrow 1$ **to** $n$ **do**
2     Initialise random $\mathbf{p}_i, \mathbf{v}_i \in \mathbb{R}^2$
3 **end**
4 **for** $t \leftarrow 1$ **to** $T$ **do**
5     Compute neighbours
6     **for** $i \leftarrow 1$ **to** $n$ **do**
7        **if** $\mathbf{p}_i$ *is close to boundary* **then**
8           $\mathbf{a}_b \leftarrow$ Steer away from boundary
9        **end**
10        $\mathbf{a}_s \leftarrow$ Avoid collisions with $\mathcal{N}(i)$
11        $\mathbf{a}_a \leftarrow$ Align to velocity of $\mathcal{N}(i)$
12        $\mathbf{a}_c \leftarrow$ Align to position of $\mathcal{N}(i)$
13        $\mathbf{a}_i \leftarrow \mathbf{a}_b + \mathbf{a}_s + \mathbf{a}_a + \mathbf{a}_c$
14        $\mathbf{v}_i \leftarrow \mathbf{v}_i + \mathbf{a}_i$
15        $\mathbf{v}_i \leftarrow$ Enforce speed and turn limits
16        $\mathbf{p}_i \leftarrow \mathbf{p}_i + \mathbf{v}_i$
17     **end**
18 **end**

**Training**  To train the GNCA we generate trajectories by letting 100 boids evolve for 500 steps, starting from randomly initialised positions and velocities. We generate 300 trajectories for training, 30 for validation and early stopping, and 30 for testing the final performance of the GNCA. The model is trained, like in the previous case, to approximate the true transition $\tau(\mathbf{S})$.

**Results**  One key aspect of approximating a continuous-state GCA is that errors in the prediction will quickly accumulate, making it almost impossible for the GNCA to perfectly imitate the true system. For this reason, despite quickly reaching a training, validation, and test error in the order of $10^{-7}$, the GNCA cannot perfectly approximate

Table 1: Complexity measures for the true Boids system and the trained GNCA.

|        | **SampEn**          | **Corr. dim.**       |
|--------|---------------------|----------------------|
| True   | $0.0369_{\pm 0.0053}$ | $1.3125_{\pm 0.1444}$ |
| GNCA   | $0.0302_{\pm 0.0079}$ | $1.3017_{\pm 0.1336}$ |

the true trajectory of the target GCA when evolved autonomously. However, we can evaluate the quality of the learned transition function by using the sample entropy (SampEn) [28] and correlation dimension (CD) [29], two measures of complexity for real-valued time series. Table 1 compares the SampEn and CD of the true Boids system with the trained GNCA, computed from trajectories of 1000 steps and averaged over all boids and 5 random initialisations of the system. We see that both systems have very similar complexity measures, indicating that their behaviour is generating a comparable amount of information and is therefore similar. We report in Fig. 4b an example of how the SampEn and CD change as the GNCA is trained. Additionally, we see that the GNCA learns a flocking behaviour similar to the target system (shown in Fig. 4c), although with smoother and less precise trajectories. Notably, we observed that the GNCA has no trouble in learning the soft-boundary condition. We include videos of the GNCA trajectories as supplementary material.

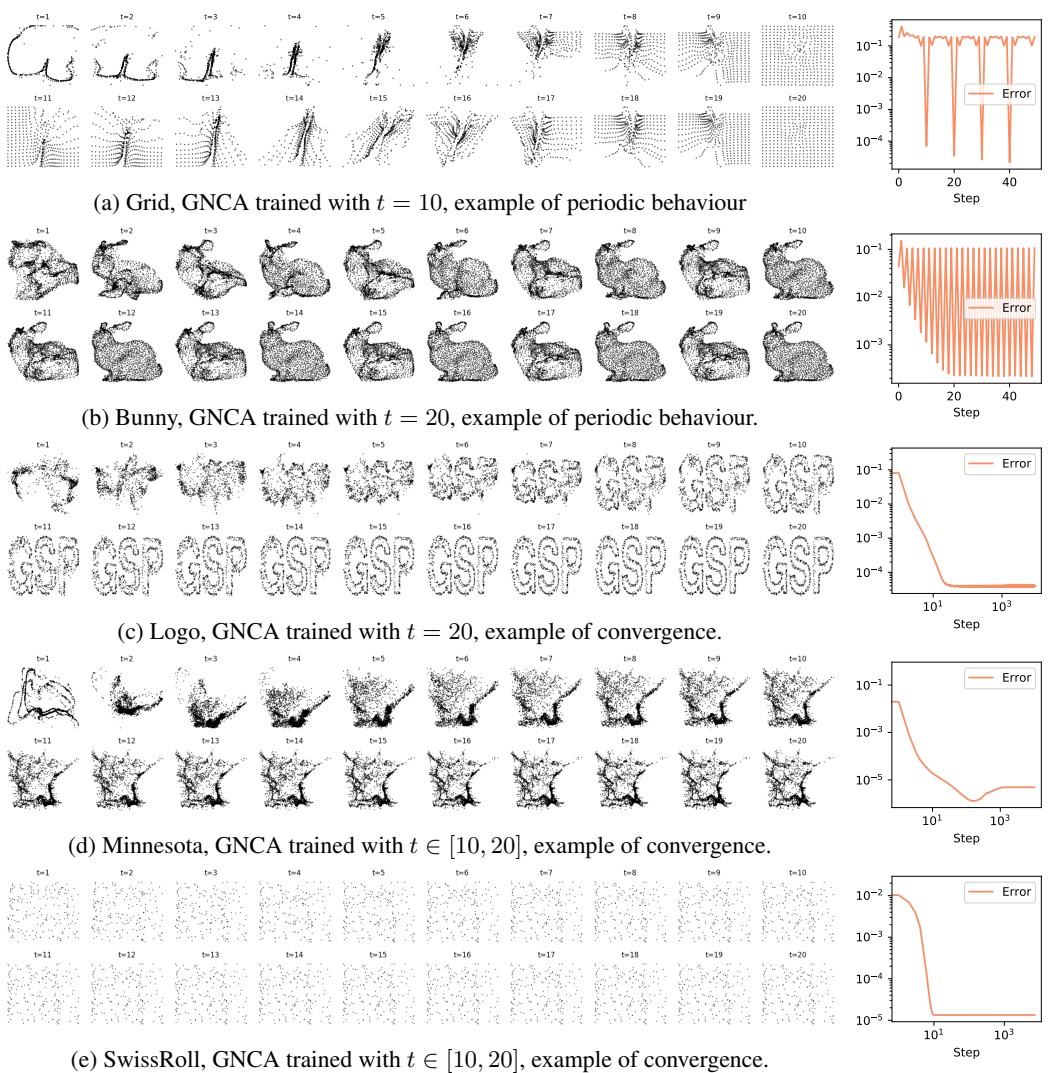

(a) Grid, GNCA trained with $t = 10$, example of periodic behaviour

(b) Bunny, GNCA trained with $t = 20$, example of periodic behaviour.

(c) Logo, GNCA trained with $t = 20$, example of convergence.

(d) Minnesota, GNCA trained with $t \in [10, 20]$, example of convergence.

(e) SwissRoll, GNCA trained with $t \in [10, 20]$, example of convergence.

Figure 5: Each row shows the state trajectory of the trained GNCA for 20 steps, starting from $\bar{\mathbf{S}}$. The plots to the right show the mean squared error between the current state and the target. A complete list of figures for all graphs and $t$ is reported in the supplementary material.

## 4.3 GNCA that converge to a fixed target

For our final experiment, we extend the previous work of Mordvintsev et al. [17] and study a setting in which a GNCA is trained to converge to a specific target state from a given initial condition. We consider several geometric graphs available in the PyGSP library [30] (BSD 3-Clause license), and we set the target state to be the 2D or 3D node coordinates, while $\mathcal{G}$ is kept fixed.

The goal of this experiment is twofold. First, we wish to study the setting in which the transition function that we are trying to approximate is unknown. Second, and more importantly, our goal is to show that the GNCA can learn local transition rules that result in a global coordination of the cells.

**Training** The main challenge in this experiment is to make the target state an attractor of the GNCA. In other words, let $\bar{\mathbf{S}}, \hat{\mathbf{S}} \in \mathbb{R}^{n \times d}$ be the initial and target states, respectively, we wish to learn $\tau_\theta$ s.t. $\tau_\theta^t(\bar{\mathbf{S}}) \to \hat{\mathbf{S}}$. Here, we consider as initial state a normalisation of the target s.t. $\bar{\mathbf{s}}_i = \hat{\mathbf{s}}_i / \|\hat{\mathbf{s}}_i\|$. Other normalisations or random positions are also possible. Note also that, for simplicity, we assume a 1-to-1 correspondence between each node and its target state.

Table 2: Type of attractor and minimum error reached by the trained GNCA, trained with different values for $t$.

| | Grid2d | | Bunny | | Minnesota | | Logo | | SwissRoll | |
|---|---|---|---|---|---|---|---|---|---|---|
| | **Type** | **Error** | **Type** | **Error** | **Type** | **Error** | **Type** | **Error** | **Type** | **Error** |
| $t = 10$ | Periodic | $1.52 \cdot 10^{-5}$ | Periodic | $4.19 \cdot 10^{-5}$ | Converge | $1.26 \cdot 10^{-6}$ | Periodic | $7.35 \cdot 10^{-6}$ | Periodic | $1.09 \cdot 10^{-5}$ |
| $t = 20$ | Periodic | $3.66 \cdot 10^{-6}$ | Periodic | $2.18 \cdot 10^{-4}$ | Converge | $1.72 \cdot 10^{-6}$ | Converge | $3.75 \cdot 10^{-5}$ | Converge | $6.26 \cdot 10^{-6}$ |
| $t \in [10, 20]$ | Converge | $1.19 \cdot 10^{-6}$ | Converge | $9.76 \cdot 10^{-5}$ | Converge | $1.29 \cdot 10^{-6}$ | Converge | $2.49 \cdot 10^{-4}$ | Converge | $1.33 \cdot 10^{-5}$ |

To train the GNCA, we apply the transition for a given number of steps $t$ and use backpropagation through time (BPTT) to update the weights, with loss $L = \frac{1}{K} \sum_{k=1}^{K} (\tau_\theta^t(\mathbf{S}^{(k)}) - \hat{\mathbf{S}})^2$ for mini-batches of size $K$ consisting of states $\mathbf{S}^{(k)}$ for $k = 1, \ldots, K$. This ensures that the GNCA will learn to converge to the target state in $t$ steps. Second, during training, we use a cache to store the states $\tau_\theta^t(\mathbf{S}^{(k)})$ reached by the GNCA after each forward pass. Then, we use the cache as a replay memory and train the GNCA on batches of states $\mathbf{S}^{(k)}$ sampled from the cache. For every batch, the cache is updated with the new states reached by the GNCA after $t$ steps, and one element of the cache is replaced with $\bar{\mathbf{S}}$ to avoid catastrophic forgetting. The cache has a size of 1024 states and is initialised entirely with $\bar{\mathbf{S}}$. By using the cache, the GNCA is trained also on states that result from a repeated application of the transition function. This strategy encourages the GNCA to remain at the target state after reaching it, while also ensuring an adequate exploration of the state space during training.

**Results** We run our experiments on the following graphs from PyGSP: a regular 2D grid, the Stanford bunny geometric mesh, the Minnesota road network, the PyGSP logo, and a random point cloud sampled on the Swiss roll manifold. Fig. 5 shows some of the state trajectories of the trained GNCA on different graphs, as well as the mean squared error between the current state and the target. We report results for $t = 10$, $t = 20$, and $t \in [10, 20]$ randomly sampled at each forward pass during training, as also done by Mordvintsev et al. [17]. The specific values that we considered for $t$ are a trade-off between computational complexity, stability during training, and a sufficient amount of time steps to allow the model to learn interesting dynamical patterns. Other values for $t$ are possible but we leave this exploration as future work.

We see that the GNCA is always able to reach a state very close to the target in the prescribed number of steps, and we highlight two main patterns that emerge by analysing the behaviour of the GNCA on different graphs and values of $t$. For $t = 10$, rather than remaining stable at the target, the GNCA learns to produce a periodic behaviour that oscillates between high-error and low-error states (as shown in Fig. 5). While this behaviour can be partially ascribed to the use of the cache, which results in backpropagating the error only after $k \cdot t$ steps for $k \in \mathbb{N}$, we see that the behaviour learned by the GNCA does not always have a period of 10 (*e.g.*, see Fig. 5b). Also, the error of the GNCA does not reach its minimum right away, but takes a few repetitions to converge to a more stable periodic orbit. We observed this behaviour also for Grid and Bunny with $t = 20$.

For $t = 20$ and $t \in [10, 20]$, the GNCA learns to converge to a stable attractor. This is expected for $t \in [10, 20]$, since the GNCA is trained to always reach the target state after any random amount of transitions. However, it is remarkable that we observe a convergent behaviour also for $t = 20$, which seemingly points to some intrinsic differences between the kind of transition rules that best approximate different graphs (since, for $t = 20$, the GNCA still learns a periodic attractor for Grid and Bunny). A summary of the behaviour of the GNCA on different graphs is reported in Table 2, as well as the minimum error reached by the trained GNCA in its trajectory.

# 5   Conclusions

We believe to have opened the way to a new line of research in which GNNs can unlock the power of GCA and implement a desired behaviour through distributed and emergent computation. The possible applications of GNCA are clear in control problems, although we believe that many complex dynamical systems can be modelled as a GCA (*e.g.*, social, epidemiological, and technological networks). In general, GNCA allow us to learn transition rules on these systems instead of hand-designing them, and could lead to a better understanding of their characteristics (*e.g.*, how a virus spreads). More research is required to address these issues since GNCA are currently not designed to be interpretable. Also, applying BPTT with GNNs can be expensive if the graph is big or dense. This is a general limitation of deep GNNs that affects our model too, and could be addressed as the field advances.

## Acknowledgments and Disclosure of Funding

This research was funded by the Swiss National Science Foundation project 200021 172671 "ALPS-FORT".

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
