# OpenReview forum: "Learning Graph Cellular Automata"
_NeurIPS.cc/2021/Conference — NeurIPS 2021 Poster_

### Official Review · Reviewer_bL59 · 2021-07-16

**Rating:** 8
**Confidence:** 3

**Summary:**

The paper presents the Graph Neural Cellular Automata (GNCA) architecture, a general-purpose architecture that uses a Graph Neural Network (GNN) to learn a desired Graph Cellular Automata (GCA). The proposed architecture is used to learn different transition rules in different kinds of space (discrete and continuous). In the paper, there is a brief (but complete) introduction to Graph Cellular Automata and Graph Neural Network. The GNCA architecture borns by the natural and clearly explained connection between GNN and GCA. The effectiveness of this architecture is demonstrated by three experiments that explore different scenarios: learning a transition rule via supervised learning in a discrete state space, learning a transition rule in a continuous state space, and learning an unknown transition rule to reproduce a target. Each result is discussed using different metrics in order to evaluate how well the model performs in that particular setting.

**Limitations And Societal Impact:**

In general, the authors did not perform an in-depth analysis of the limitations of their work. For example, it would be interesting to see some experiments where the architecture does some errors, maybe trying different agent-based contexts (different from the one proposed in section 4.2). It would be interesting to see more than one experiment for each context (this has been done only for the convergence to a fixed target, section 4.3). In this way, we know that the architecture works pretty well only in the proposed experiments. However, for each experiment, when a particular behavior occurs, it has been described/investigated very well by the authors (an example is the periodic behavior explained starting from line 235).


**Main Review:**

* The paper is complete, well written, and clear in the explanation of the concepts;
* the experiments are various and describe well the effectiveness of the architecture in different settings. A great contribution is given by the description in the supplementary material;
* the paper proposes an interesting and simple architecture to learn transition rules in a more general setting with respect to previous works in the field;
* some particular results from the experiments are highlighted and well-argued (e.g. the periodic behavior on figure 5).

As said above, the paper is well written and everything is defined precisely so there are no relevant weaknesses, only some clarifications:
* Even if well described, it would be more clear if the model description in appendix's section 1.2 were moved in the paper, in section 3.3;
* algorithm 1 is not clear in some of its points: at line 2, $p_i$ and $v_i$ are not clearly defined. I suppose they refer to the position and velocity of boid $i$, but at line 46 in the appendix, you use $p_x, p_y$ for position and $v_x,v_y$ for velocity, so this is a little bit confusing. Moreover, lines 10,...,16 (of algorithm 1) are not very clear;
* there are no baselines. If the architecture had been compared with some existing methods we would have a more clear way to understand its validity;
* a more in-depth explanation of the metrics used in the experiments would be interesting (e.g. SamplEn), maybe using some formulas. This could also be done in the appendix;
* neither in the paper nor in the appendix there is a specification on how you chose the hyperparameters;
* at line 301, what kind of normalization did you use? Have you tried different kinds of normalization? Why not try using random positions? Each starting point has a corresponding target point, is possible that two or more points exchange their target point? Because in your setting this would be considered as an error (that will increase the error). For example, if at the end $\bar{s}_i = \hat{s}_j$ and $\bar{s}_j = \hat{s}_i$ with $i\neq j$ this is an error for you but the target is correctly reconstructed. I have understood your idea but it will be interesting if this were analyzed or only mentioned;
* line 304 is not clear to me. Each mini-batch has size $K$ but you mention that each of them consists of initial states $S^{(k)}=1,\dots, K$. This is misleading because it appears that each mini-batch is composed of the same points;
* in the conclusion section, it would be interesting to read a "future work" analysis, describing how the work could continue (this has been done in some parts of the paper, but it would be more clear to have a dedicated section), and with a brief discussion about the limitations of the proposed architecture.

Minor typos and errors:
* In the code (README.md), you mention folder "point_cloud" but this does not exist in the code. I suppose that the correct name is "fixed_target";
* at line 290 you refer to a video of the GNCA trajectories in the supplementary material, but there is not.


**Time Spent Reviewing:**

9

---

> ### Author Response · Authors · 2021-08-08
> **Thanks for the positive feedback, we have addressed all suggestions**
>
> We thank the reviewer for the exceptionally positive feedback, for recognizing the value of our work, and for the many suggestions for improving our paper. We also thank the reviewer for the many hours dedicated to reviewing our paper. We have addressed all of the reviewer's comments, and we report a summary of changes below.
>
> **Summary of changes concerning clarity, minor errors, etc.**
>
> - We have improved Algorithm 1 and fixed the inconsistencies in the notation.
> - We have added a description of the two complexity measures.
> - We have added the description of how hyperparameters were chosen.
> - We have improved the description at line 304: states are sampled randomly from the cache for each mini-batch, they are not necessarily all equal.
> - We have renamed the `fixed_target` folder.
> - We apologize for the missing videos, we forgot to include them in the zip file. We will update the supplementary material.
>
> **Other changes in response to the comments of the reviewer**
>
> > If the architecture had been compared with some existing methods we would have a more clear way to understand its validity.
>
> We could not identify potential baselines, because they would need to essentially be a GNCA themselves in order to work in the various experiments.
>
> For example, all methods discussed in Sec. 2 can operate only on regular grids. Similarly, MLPs cannot model edge relations at all. Using other kinds of GNNs to implement a GNCA is an interesting possibility, but we leave it as future work concerning the design of GNCA for specific problems.
>
> We have added this consideration to the paper.
>
> > What kind of normalization did you use? Have you tried different kinds of normalization? Why not try using random positions?
>
> We normalize all states by their norm, as described in line 301.
> Other normalizations and random positions are a possibility. For example, we show in experiments 1 and 2 that GNCA can deal with random states quite well.
>
> We have added this consideration to the paper.
>
> > Is possible that two or more points exchange their target point?
>
> In general, yes, although it depends on the specific setting.
>
> If the graph is static then the target of each node might depend on the connectivity, e.g., if we switch two nodes in a point cloud we might “tangle” the mesh. If the graph is dynamic then nodes are likely to be interchangeable and what the reviewer says is definitely a possibility.
>
> We have added this consideration to the paper, as suggested by the reviewer.
>
> > It would be interesting to read a "future work" analysis [... in] a dedicated section
>
> Thanks for the suggestions, we have rearranged and expanded the discussion on future works. A brief excerpt from the paper (also, see next comment):
>
> _“The possible applications of GNCA are clear in control problems, although we believe that many complex networked dynamical systems can be modeled as a GCA (e.g. social networks, epidemiological networks, and many technological networks). In general, GNCA allows us to learn transition rules on these complex systems without designing them a priori, and could lead to a better understanding of their characteristics (e.g., how does a virus spread?).”_
>
> > Limitations And Societal Impact
>
> Thanks for the suggestion, we have improved the discussion of the limitations of the method.
>
> The biggest limitation is the poor interpretability of the learned rules (e.g., what is the difference between a converging and an oscillating GNCA?). This issue could be addressed through explainability techniques, stability analysis, etc., but it requires dedicated research efforts.
>
> Additionally, applying BPTT with GNNs can be very memory-expensive if the graph is big or the adjacency matrix is dense. This is a limitation of any deep GNN, that affects our model as well.
>
> We agree with the reviewer that more experiments, including negative results, would also be interesting to see. For this paper, we have focused on experiments that represent specific scenarios so that we could perform an in-depth analysis. We hope that future works will build on our paper to explore other scenarios.

---

> ### Comment · Reviewer_bL59 · 2021-08-23
> **Comment after authors' response**
>
> The authors provided in-depth and complete answers to each question/doubt expressed in the review. Particularly, the "future work" and "limitations" sections are important additions for the paper's completeness. The evaluation remains unchanged.

---

### Official Review · Reviewer_ZxB8 · 2021-07-17

**Rating:** 6
**Confidence:** 5

**Summary:**

Authors present a general-purpose architecture for learning GCA, and we show that it can represent any arbitrary GCA with finite and discrete state space. Then, we test our approach on three different tasks: 1) learning the transition rule of a GCA on a Voronoi tessellation;
2) imitating the behaviour of a group of flocking agents; 3) learning a rule that converges to a desired target state.

**Limitations And Societal Impact:**

Please addressed the limitations and potential negative societal impact of your work.

**Main Review:**

Authors have proposed a general-purpose architecture to achieve this, which we showed to be general enough to implement
any arbitrary transition function for a large class of GCA. Authors have applied the GNCA model to learn a variety of GCA transition rules, including unknown rules for which we only specified a target state.

**Time Spent Reviewing:**

3

---

> ### Author Response · Authors · 2021-08-08
> **Thanks for the positive rating**
>
> We thank the reviewer for the positive assessment of our paper and we have addressed the limitations and potential negative societal impact of our work.
> We remain available to answer any questions or provide clarifications.

---

### Official Review · Reviewer_Tp3K · 2021-07-17

**Rating:** 6
**Confidence:** 4

**Summary:**

This paper presents an approach to learn, what the authors call, graph cellular automata (GCAs). Graph neural networks are trained to learn transition rules for  GCAs and tested on three different domains: Voronoi tessellation, learning dynamics of flocking agents, and learning to approximate 2D/3D target structures. The main contribution is the application of graph neural networks to these three interesting problems.


**Limitations And Societal Impact:**

The potential societal impacts are not elaborate on. It could be worthwhile to elaborate a bit on the dangers and opportunities in systems that are based on the self-organization of locally-communicated parts. These systems might be even harder to control and impossible to predict.

**Main Review:**

The paper is in general well written and clear. It also addresses an interesting and important topic, which is how can we train systems whose global behaviors arise purely through the local interaction of cells. Additionally, the three chosen domains show the diversity of behaviors that are learnable by a graph neural network. However, why exactly these three domains are chosen could be elaborated on and also what interesting future work they enable.

My main critique is that it’s not very clear how exactly the current system differs from a graph neural network. Isn’t a neural cellular automata a subset of a graph neural network already? It would be great if the authors could be more precise here on how exactly their system differs from a graph neural network or how both are defined exactly. For example, in the abstract, the authors note that: “...we use graph neural network to learn a variety of transition rules for GCA”. So in this case the GNN is regarded as the learning algorithm? Here it might also help to clearly state the unique contributions in the introduction of the paper. Currently, the paper looks more like “just” and application for GNN to three interesting problems.

I wasn’t sure what the motivation is behind the minimal exact implementation. Some motivation would help the reader here.

There is some work from the evolutionary computation community to learn transition rules for cellular automata with neural networks, which predates the approaches mentioned in this paper. For example:

Evolving Self-organizing Cellular Automata Based on Neural Network Genotypes. Elmenreich, W. and Fehervari, I., 2011. Self-Organizing Systems, pp. 16--25. Springer Berlin Heidelberg.

CA-NEAT: Evolved Compositional Pattern Producing Networks for Cellular Automata Morphogenesis and Replication. Nichele, S., Ose, M.B., Risi, S. and Tufte, G., 2018. IEEE Transactions on Cognitive and Developmental Systems, Vol 10, pp. 687-700.

Learning Cellular Automaton Dynamics with Neural Networks. Wulff, N.H. and Hertz, J.A., 1992.  Neural Information Processing Systems, pp. 631–638.



**Time Spent Reviewing:**

4

---

> ### Author Response · Authors · 2021-08-08
> **Thanks for the overall positive assessment. We clarify the reviewer’s doubts.**
>
> We thank the reviewer for recognizing that our work addresses an interesting and important topic, and for the overall positive assessment.
>
> Please allow us to clarify some aspects that we feel are due to simple misunderstandings, especially concerning the main critique.
>
> **Regarding the main critique**
>
> > My main critique is that it’s not very clear how exactly the current system differs from a graph neural network.
>
> A Graph Neural Cellular Automaton (GNCA) is a Graph Cellular Automaton (GCA) in which the transition function is implemented and/or learned with a Graph Neural Network (GNN).
>
> Our goal isn’t to present something different from a GNN, but to show that GNNs are a natural choice for implementing GCA transition functions. Crucially, they enable learning the transition function instead of hand-designing it. We discuss this in Section 3.3, comparing the formal definitions of GCA and GNNs given in Sections 3.1 and 3.2.
>
> Our main contributions are the following:
>
> 1. We discuss a GNN architecture that can provably implement any arbitrary transition function for GCA.
> We show that all possible transition functions of GCA can be implemented with this architecture and how we can use this to our advantage.
>
> 2. We show applications of our GNCA to learn the transition functions of different types of GCA (experiments 1 and 2) and even learn an unknown transition function of which we only specify the high-level behavior (experiment 3).
>
> 3. We also show that, in some cases, a GNCA can implement a given rule using a very small number of parameters, as in the minimal exact implementation in 4.1 (this also responds to the other comment of the reviewer).
>
> We have added this list to the introduction, we thank the reviewer for the suggestion.
>
> > Isn’t a neural cellular automata a subset of a graph neural network already?
>
> NCA are a subset of GNCA, because they are based on CNNs which are a subset of GNNs.
>
> NCA cannot be used to learn rules on arbitrary graphs, they are limited to images or other static regular grids. Instead, GNCA are a more general approach that can deal with all these cases.
>
> Also, since NCA are a subset of GNCA, our theoretical analysis holds for NCA as well.
>
> **Regarding the applications**
>
> > Why exactly these three domains are chosen could be elaborated on.
>
> The three experiments represent three different, yet important scenarios.
>
> 1. Discrete states, fixed graph: this is the default GCA case and the simplest extension of a typical CA.
> 2. Continuous states, dynamic graph: this is the most general GCA case. The experiment represents a typical real-world scenario where the GCA describes a multi-agent system.
> 3. Finally, we test the case in which we want to design a transition function. While manually designing the function would have been extremely difficult, we can specify its desired behavior --e.g., converging to a target-- and then learn it with the GNCA,
>
> We have re-worded the first paragraph of Section 4 to highlight these considerations better.
>
> >  The paper looks more like “just” and application for GNN to three interesting problems.
>
> The experiments are an interesting and important part of our paper, but they also support our main contributions in non-trivial ways.
>
> First, they show that GNCA are successful in very diverse settings (discrete/continuous states, static/dynamic graphs). This provides empirical confirmation of our theoretical analysis.
>
> Second, they show a very important application of GNCA: even if we don’t know how to design a GCA that converges to, e.g., a bunny point cloud, with our method we can specify the target and let the GNCA learn the correct rule. This enables a lot of practical applications in the future.
>
> **Other suggestions by the reviewer**
>
> > There is some work from the evolutionary computation community to learn transition rules for cellular automata with neural networks.
>
> We thank the reviewer for pointing out these works to us. We have included them in the relevant section, where we discuss previous architectures for learning CA rules.
>
> We recall again how our GNCA is more general than previous approaches since it also works for regular grids as a particular case.
>
> > Limitations And Societal Impact
>
> We thank the reviewer for the suggestion, we have included these considerations in the paper.

---

> > ### Comment · Reviewer_Tp3K · 2021-08-18
> > **Thank you for the clarifications**
> >
> > Thank you for your answers to my questions. The contributions of the paper are more clear to me now and I increased my score from 5 to 6.

---

### Decision · Program_Chairs · 2021-09-27

**Decision:**

Accept (Poster)

**Comment:**

Meta-review of "Learning Graph Cellular Automata"

This paper applies graph neural networks to learn transition rules of cellular automata. They demonstrate results on the proposed graph cellular automata (GCA) method on 3 different domains that are popular with CA-based approaches: Voronoi tessellation, learning dynamics of flocking agents, and learning to approximate 2D/3D target structures.

All reviewers agree that the paper is well written and explains concepts clearly, with particular emphasis on details for all the well-chosen experiments. The results further highlight how interactions of local cells can produce global coherent phenomena.

Reviewer Tp3K notes that there has been much work in the evolutionary computation community about learning transition rules for CA with neural nets, and suggested a few, so I highly recommend the authors update the work with a discussion on historical works. This is important, because this paper can help guide the NeurIPS community into CA's, and one must provide the right context to a mature field that has existed for a long time.

For all these reasons stated, I recommend acceptance of this work.